# Vaccination Knowledge, Attitudes and Practices Among Healthcare Students in Spain: Development and Psychometric Validation of a Life-Course Immunization Questionnaire

**DOI:** 10.3390/vaccines14010009

**Published:** 2025-12-20

**Authors:** Magdalena Santana-Armas, Olalla Vazquez-Cancela, Isabel Ferreiro-Cadahía, Cristina Peiteado-Romay, Daniel Lorenzo-Fuente, Cristina Fernández-Pérez, Juan Manuel Vazquez-Lago

**Affiliations:** 1Department of Preventive Medicine, Santiago de Compostela University Teaching Hospital, 15703 Santiago de Compostela, Spain; 2Health Research Institute of Santiago de Compostela (IDIS), 15706 Santiago de Compostela, Spain; 3Department of Preventive Medicine and Public Health, University of Santiago de Compostela, 15786 Santiago de Compostela, Spain

**Keywords:** questionnaire validation, attitudes, knowledge, healthcare students, vaccination

## Abstract

Vaccine hesitancy represents a threat to immunization programs and herd immunity. Our objective was to validate a Spanish-language questionnaire to assess knowledge, attitudes, and practices (KAP) of students in the healthcare field regarding vaccination and the immunization schedule. Methods: An online questionnaire was developed and distributed via RedCap v.13.7.1 to healthcare students undertaking clinical placements at the University Hospital of Santiago de Compostela during the 2024–2025 academic year. The questionnaire assessed nine dimensions through thirty-four items. Validation was carried out in two phases: (1) translation and expert content validation, and (2) reliability testing using Cronbach’s alpha and validity assessment through principal component analysis (PCA). Results: A total of 398 students completed the questionnaire. The mean age was 23.78 ± 3.77 years. Of these, 19.60% were men (n = 80) and 77.50% were women (n = 316). Validation of the questionnaire was carried out with a random sample of 294 students. The final 30-item questionnaire demonstrated high internal consistency (Cronbach’s alpha = 0.83) and construct validity, confirmed by PCA, supporting the presence of nine dimensions that explained 60.93% of the total variance. Overall, 74.70% of students reported that scientific evidence was the main influence on their opinion about vaccines. Regarding practices, 76.10% believed that certain vaccines should be mandatory for healthcare personnel. Conclusions: The questionnaire demonstrated reliability and validity for evaluating KAP on vaccination among future healthcare professionals. Having this instrument available will help guide future educational interventions and strengthen their role as trusted agents in immunization.

## 1. Introduction

Vaccination is one of the most useful and cost-effective interventions for the prevention of infectious diseases, contributing decisively to the improvement of survival and the reduction in disease burden worldwide [1,2]. The development of immunization programs has made it possible to control and even eradicate certain diseases, consolidating vaccination as a cornerstone of public health [3]. However, vaccine hesitancy has been recognized by the World Health Organization (WHO) as one of the main health threats of the 21st century, as it jeopardizes the success of vaccination programs and compromises herd immunity [3,4]. In the general population, vaccine acceptance depends on multiple factors, among which risk perception, misinformation, exposure to falsehoods on social media, and various cultural, cognitive, and structural barriers stand out [5,6]. These circumstances may contribute to low vaccination coverage, increasing vulnerability to epidemic outbreaks [7,8]. These challenges have become particularly evident in the 21st century, in the wake of the COVID-19 pandemic, the rapid development and deployment of novel vaccine platforms, and the amplification of vaccine-related misinformation through digital and social media, all of which have contributed to vaccine hesitancy and undermined confidence in vaccination programmes [9,10,11,12,13].

In this context, the analysis of knowledge, attitudes, and practices (KAP) toward vaccination has become a key tool for identifying barriers and designing strategies aimed at improving vaccine confidence. Healthcare professionals occupy a central role in this process, as they are perceived as a trustworthy source of information and play a crucial role in promoting vaccination [14,15]. Consequently, it is essential that future healthcare professionals receive appropriate training during their formative years, enabling them to acquire solid competencies in immunization. Nevertheless, different studies conducted in Europe and other regions have identified gaps in the knowledge and attitudes of healthcare students regarding vaccination, as well as variability in vaccine coverage within this group [14,15].

Despite the relevance of the topic, there is no validated questionnaire available in Spanish that reliably assesses the knowledge and attitudes of healthcare students toward vaccination, while also including specific vaccines covered by national immunization programs. Having an adapted and validated instrument is essential to obtain consistent data that allow the identification of areas for improvement, guide educational interventions, and strengthen the role of future professionals as trusted agents in immunization.

The main objective of this study was to validate a Spanish-language questionnaire for assessing the knowledge and attitudes of healthcare students regarding vaccination and their vaccination schedule. As a secondary objective, we aimed to describe the results obtained from the application of the questionnaire in a sample of healthcare students undertaking hospital internships at the University Hospital of Santiago de Compostela during the 2024–2025 academic year.

## 2. Materials and Methods

A validation study was conducted on a questionnaire assessing knowledge and attitudes toward vaccination, aimed at students enrolled in healthcare-related degrees who completed hospital internships during the 2024–2025 academic year. The study was carried out between October 2024 and May 2025 at the University Clinical Hospital of Santiago de Compostela, a city located in the northwestern region of Galicia, Spain. Subsequently, a descriptive cross-sectional study of the responses obtained was performed.

A previous European study conducted with medical students [15] was used as a reference for the design and adaptation of the questionnaire. The instrument includes both: (1) vaccines from the routine national immunization schedule, and (2) certain vaccines recommended in specific epidemiological, travel, or occupational circumstances. Modifications were made to tailor it to the target population and its vaccination schedule. The adapted questionnaire was then validated to ensure its suitability for the local context.

The KAP (Knowledge, Attitudes, and Practices) framework guided the categorization of the questionnaire dimensions. Specifically, dimensions classified as “knowledge” captured objective and factual information, whereas “attitudes and perceptions” reflected beliefs and subjective opinions. KAP-related variables were grouped into three blocks: (1) the knowledge block, includes questions on awareness and understanding of different vaccines (childhood, special-risk, respiratory, hepatitis, influenza, and newer vaccines); (2) the attitudes block includes items on confidence in vaccines, perceived importance and safety, and views on vaccine mandates; and (3) the practices block includes self-reported vaccination status for each vaccine and awareness of booster/ revaccination needs. Each studied dimension corresponded to a block of questions, and a fourth block evaluated demographic aspects, which were not included in the validation process.

The demographic block included open-ended questions on year of birth, country of study, degree program, type of clinical placement or setting and the academic year. A closed-ended question on gender offered three options: female, male, or other. Participants who did not identify as male or female, or preferred not to disclose their gender, could select “other.” The remaining blocks contained single-choice closed-ended questions, except for the “Other vaccines” question, which was open-ended and allowed a free-text response.

Students enrolled in healthcare-related programs were invited to participate if they met one of the following criteria:(i)attending a medical examination prior to the start of hospital internships during the 2024–2025 academic year, or(ii)enrolled in the Preventive Medicine course in the Medical Degree program at the University of Santiago de Compostela who attended class.

Only students who voluntarily completed and submitted the anonymous questionnaire were included in the study.

The invitation to participate was issued during the medical examination prior to the start of hospital internships and during in-person sessions of the Preventive Medicine course taught in the fifth year of the Medical Degree. Participation was voluntary, and no incentives were offered.

Access to the online questionnaire was provided through a QR code. Data collection was carried out via the RedCap platform (Vanderbilt University, Nashville, TN, USA) [16] from early October 2024 to late April 2025.

### 2.1. Validation and Reliability of the Questionnaire

For questionnaire validation, a random sample was selected using the EpiDat 4.2 software, yielding 294 questionnaires. The process was conducted in two phases (Figure 1).

Step 1. Translation and content validation of the questionnaire.

The adaptation of the questionnaire into Spanish followed the standardized WHO-recommended translation and validation process [17], which included:(i)forward translation by two independent translators;(ii)synthesis of both translations through consensus; and(iii)back-translation by another independent translator to verify content fidelity.

Subsequently, content validity was assessed by a nominal group of experts from the Department of Preventive Medicine and Public Health of the Clinical Hospital of Santiago de Compostela. The expert panel, experienced in questionnaire design, evaluated aspects such as grammar, syntax, organization, appropriateness, and coherence of the items. All versions were reviewed to ensure conceptual and cultural equivalence, assess content accuracy, and achieve consensus on item comprehension in Spanish.

Step 2. Reliability analysis.

Reliability analysis was conducted in two stages under the framework of Classical Test Theory (CTT) [18]:(i)evaluation of internal consistency through Cronbach’s alpha coefficient and item discrimination analysis. The homogeneity index was determined using Pearson’s correlation coefficient between the item score and the sum of the remaining item scores. Items with a homogeneity index below 0.20 were eliminated, as they did not measure the same construct as the rest of the items;(ii)evaluation of the resulting model using Principal Component Analysis (PCA) with Varimax rotation. The Kaiser criterion was applied to select components, verifying the robustness of the results and considering relevant those with at least two items showing correlations >0.40. As a sensitivity analysis, PCA with Promax rotation (κ = 4) was also conducted. Prior to the factorial analysis, sample adequacy was assessed through the Kaiser–Meyer–Olkin (KMO) measure and complemented by Bartlett’s test of sphericity [19].

Data analysis was performed using Microsoft Excel and SPSS software version 29.0.1.0 (IBM, Armonk, NY, USA).

After validation of the questionnaire, a descriptive analysis of the responses was carried out to explore students’ knowledge, attitudes, and practices regarding vaccination.

### 2.2. Statistical Analysis

A descriptive statistical analysis was performed to characterize the sample. Participants were grouped according to their degree program, one group for Medicine students and another combining the remaining degrees: Nursing, Nursing Care Assistants Technicians (NCAT), Dentistry, Laboratory Technicians, Imaging and Radiology Technicians, and Radiotherapy Technicians. Participants were also grouped by academic year, comparing fifth-year students with those in other years, regardless of their degree program.

Results were analyzed by gender, degree, and academic year. Participants identifying as “other gender” were excluded from gender-based comparisons but were included in analyses by degree, academic year, and in the overall analysis. Data analysis was conducted using Microsoft Excel and SPSS software version 29.0.1.0 (IBM, Armonk, NY, USA).

All analyzed variables were categorical. Counts (n) and percentages (%) of responses were estimated for each question, considering the total sample and subgroups by gender, degree, and academic year. Since some contingency table cells were expected to have counts below 5, Fisher’s exact test was used to compare responses between men and women. Statistical significance was established at *p* < 0.05.

### 2.3. Ethical Considerations

Participation was voluntary and anonymous. By completing the questionnaire, students implicitly consented to participate in the research project; therefore, no individual consent forms were collected.

No incentives were provided for participation. The study complied with the ethical principles of the Declaration of Helsinki and with current legislation on personal data protection (Regulation (EU) 2016/679) [20].

The study was approved by the Santiago–Lugo Research Ethics Committee (registry number 2022/259).

## 3. Results

### 3.1. Validation of the Questionnaire

The KAP questionnaire on vaccination was subjected to internal consistency and factorial adequacy analyses (Table 1). The preliminary version, consisting of 34 items, yielded an overall Cronbach’s alpha of 0.79, and most knowledge items related to individual vaccines showed corrected item–total correlations above 0.30, indicating good homogeneity. However, some items concerning attitudes or opinions (for example, perceptions of influenza vaccination, vaccination mandates, or vaccination during pregnancy) showed low or even negative correlations (<0.20).

The adequacy of the correlation matrix for factorial analysis was satisfactory, with a Kaiser–Meyer–Olkin (KMO) value of 0.77, while Bartlett’s test of sphericity was significant (*p* < 0.01), confirming the suitability of the correlation matrix for factorial analysis. Principal Component Analysis (PCA) identified 11 components with eigenvalues (λ) > 1, explaining 62.10% of the total variance. After Varimax rotation, the items grouped coherently into factors, reflecting an adequate initial structure. Table 2 presents each component’s contribution to the overall scale and the reliability coefficients obtained for each dimension.

Based on the homogeneity and consistency indices, four items with low correlations (<0.20) were removed: Item 6: “Which statement best describes your opinion about vaccines?” Item 7: “Do you believe vaccination programs are an effective tool for disease prevention?” Item 36: “Do you think there should be a more specific vaccination program for pregnant women (e.g., against seasonal influenza, mumps, or rubella)?” Item 37: “What is your opinion on the seasonal influenza vaccine?” This refinement improved the questionnaire’s overall homogeneity and reliability, maintaining an adequate Cronbach’s alpha for the remaining items.

The final version of the questionnaire comprised 30 items (plus demographic questions), distributed across theoretical blocks corresponding to knowledge, attitudes, and practices. The recalculated internal consistency (Table 3) was good (α = 0.83), with acceptable sampling adequacy (KMO = 0.79) and a significant Bartlett’s test, *p* < 0.01.

A new rotated component matrix (Table 4) confirmed the assignment of each item to distinct factors, yielding very similar results. Nine components now explain approximately 61% of the variance. These findings support the reliability and factorial structure of the questionnaire after item refinement.

After Varimax rotation, items were grouped into nine factors with factor loadings ≥ 0.40, enabling the identification of nine distinct dimensions. Each dimension was named according to the content of its items, following the consensus of the research team and reflecting the most reasonable conceptual description. The resulting dimensions were:Classical childhood vaccines (Items 12–16, 22)Special or less frequent vaccines (Items 25–29, 31)Respiratory vaccines (Items 19–21, 23)Hepatitis (Items 17 and 18)Varicella/schedule (Items 9, 20, 30)Influenza/seasonal (Items 11, 34, 38)Attitudes (Items 8, 33–35)New vaccines (Item 32)Mandates/obligations (Items 39 and 40)

Each factor demonstrated acceptable to good internal consistency. Subsequently, these dimensions were mapped to the three conceptual KAP domains. Dimensions addressing technical or specific vaccine-related aspects were classified as “knowledge,” those representing opinions or perceptions as “attitudes,” and those related to vaccination behavior as “practices.” The distribution was as follows:Knowledge: Factors 1, 2, 3, and 4Attitudes: Factors 7 and 9Practices: Factors 5, 6, and 8

### 3.2. Demographic Characteristics

A total of 483 individuals were invited to participate, of whom 398 completed the survey, yielding a response rate of 82.40%. No questionnaires were excluded, and since variables were analyzed independently, incomplete surveys were included in the study.

The mean age of participants was 23.78 years (range 18–56; SD = 3.77). Participant characteristics are shown in Table 5.

Among respondents, 77.50% were women, 19.60% men, and 0.50% identified as another gender. Most (93.70%) were Spanish nationals. In terms of degree, 77.00% were Medical students, followed by 15.20% in Nursing. Regarding academic year, 60.80% were in their fifth year of study.

### 3.3. Knowledge

Self-reported vaccination coverage was high for classical childhood vaccines (Table 6): tetanus (91.70%), diphtheria (75.00%), poliomyelitis (69.20%), and *Haemophilus influenzae* type B (68.90%). However, pertussis showed lower coverage (43.20%). For less common or special vaccines (Table 7), such as cholera (13.10%), typhoid (16.70%), or tuberculosis (22.50%), lack of awareness predominated (up to 46.00%). Among respiratory vaccines (Table 8), coverage was also high: measles (90.90%), mumps (76.30%), and meningococcal (83.60%), though lower for pneumococcal (68.70%).

Hepatitis vaccines (Table 9) showed varying coverage: hepatitis B (86.10%) versus hepatitis A (58.10%), with the latter showing higher levels of uncertainty. Overall, Medical students and those in advanced academic years reported higher coverage and knowledge, with statistically significant differences compared to other degrees.

### 3.4. Attitudes

Regarding attitudes (Table 10), 74.70% of participants cited scientific evidence as the main influence on their opinion about vaccines, followed by physicians and professors (23.50%). The influence of social media, family, or friends was minimal. A total of 80.40% reported having received the HPV vaccine, with significant gender differences: 90.50% among women versus 40.00% among men (*p* < 0.01). Recommendation of vaccination to family, friends, or colleagues was almost unanimous (95.70%).

Concerning mandates (Table 11), 76.10% supported making influenza and hepatitis B vaccines mandatory for healthcare professionals, while 71.40% supported mandates for medical students. In contrast, 23.00% and 28.00%, respectively, believed vaccination should remain voluntary, and none expressed safety concerns about vaccines.

### 3.5. Practices

For the “Practices” block, three factors were analyzed: Schedule (Table 12), Influenza (Table 13), and New vaccines (Table 14). A total of 99.20% of students reported compliance with their national vaccination schedule. Self-reported coverage for measles was 90.90%, while varicella reached 62.10%. Seasonal influenza vaccination rates were lower: 57.30% reported having received it at least once, but only 39.90% were vaccinated annually.

Regarding awareness of revaccination, 73.40% claimed to comply with booster recommendations, although 26.40% admitted uncertainty about their vaccination status. For new vaccines, only 20.50% reported being vaccinated against herpes zoster, while 28.00% were unaware of their status.

## 4. Discussion

The present study aimed to validate a Spanish-language questionnaire for assessing knowledge, attitudes, and practices (KAP) among students in healthcare-related degrees regarding vaccination and their immunization schedule. The results confirmed the internal consistency and construct validity of the instrument, which was structured into nine dimensions encompassing classical childhood vaccines, perceptions of new vaccines, and attitudes toward vaccination mandates. Additionally, the descriptive analysis revealed that surveyed students generally held favorable attitudes toward vaccination, demonstrated adequate confidence in scientific evidence, and showed high compliance with vaccination schedules, although certain gaps persisted in specific practices and coverage areas.

First, we validated a KAP questionnaire on vaccination and vaccination schedules among healthcare students, which, after the elimination of four low-performing items, demonstrated good internal consistency and sampling adequacy with a significant Bartlett’s test of sphericity. This confirmed the factorability of the correlation matrix and supported its suitability for measuring KAP within this population. The structure of nine components retained according to Kaiser’s criterion explained approximately 61.00% of the variance and was organized into conceptually interpretable blocks, providing evidence of construct validity consistent with previously validated vaccination attitude questionnaires in Spain [21,22]. These results confirm that the questionnaire has a solid, reliable, and coherent factorial structure that allows for differentiated assessment of knowledge, attitudes, and practices related to vaccination among healthcare students. The main innovation of our questionnaire was the inclusion of self-reported vaccination status, allowing comparisons between knowledge, attitudes, and real practices, thereby adding value compared to previously published instruments.

Our multidimensional solution in Spanish, consistent with prior research [21,22,23,24], confirms stable attitudinal domains and shows high reliability and predictive validity for vaccination intent and decision-making. This suggests that our KAP subscales fit within a well-established theoretical framework [21]. Likewise, validation of a previously developed questionnaire in another language and cultural setting, applied to healthcare students, produced a high Cronbach’s alpha value obtained through parallel analysis. This indicates that concise, well-focused scales can remain highly stable when the content is precise, complementing our broader KAP assessment approach [22].

Regarding knowledge, self-reported coverage was high for classical vaccines (tetanus, poliomyelitis, diphtheria, rubella, measles), which is expected given the high childhood vaccination coverage in Spain and Portugal [8]. However, notable gaps were found in self-reported pertussis vaccination, where over 50% of respondents did not recall their status, even though this vaccine is typically administered together with tetanus. This finding reflects both limited vaccination knowledge and students’ difficulty recalling childhood doses. This pattern has been documented in the literature, where actual coverage tends to be higher than perceived coverage, highlighting the need to improve information systems and education on vaccination schedules [25].

International evidence among students of Medicine, Nursing, and Pharmacy shows consistent relationships between degree program and academic level regarding beliefs and vaccination intent, supporting the notion that greater knowledge fosters more favorable vaccination attitudes and, in turn, positive behaviors toward immunization. Our subscale data reinforce this pattern and position curricular education as a key lever for change [23,24]. Furthermore, in contexts with low influenza vaccination coverage, both intent and actual vaccination rates improve when structural barriers (e.g., lack of access or cost) are removed and when students perceive the vaccine as effective, which justifies educational interventions emphasizing vaccine efficacy [26].

Additionally, psychosocial determinants such as trust and perceived risk have been shown to predict vaccination intent and positive behavior among both healthcare students and the general population. This underscores the importance of incorporating measurement and intervention strategies addressing these dimensions into educational programs [27,28,29]. In Spain, during the COVID-19 pandemic, an improvement in vaccination attitudes was observed among health science students, suggesting a possible period effect and reinforcing the opportunity to institutionalize vaccination practices and evidence-based learning in university settings [30].

A particularly interesting finding concerned attitudes toward vaccine mandates: while most students supported mandatory vaccination for healthcare professionals, the level of agreement was slightly lower for medical students. This difference may be explained by the perception that practicing professionals have greater responsibility for infection prevention and patient safety compared to students still in training. Similar patterns have been described in other European studies, where endorsement of vaccination mandates increases with proximity to professional practice [23,25]. Nevertheless, the high levels of acceptance observed in both cases highlight an overall awareness of vaccination as a duty of care within healthcare professions.

Regarding practices, adherence to the vaccination schedule and coverage for measles and hepatitis B were high, but lower for seasonal influenza and certain booster vaccines. This pattern aligns with findings from previous studies showing that influenza vaccination remains suboptimal even among healthcare professionals, often due to misconceptions about its effectiveness or low perceived risk of infection [26,28]. The data also showed that 73.40% of students acknowledged the need for revaccination, although over a quarter were unsure of their vaccination status, revealing a gap between knowledge and actual behavior. This finding reinforces the need to promote occupational vaccination campaigns in hospitals and universities, providing accessible and convenient vaccination services for students in clinical training.

The high acceptance of vaccination and the prominent role of scientific sources or the academic environment as information references that we observed are consistent with Spanish and European literature and support the appropriateness of evidence-based interventions [21,22,30]. The practice gap observed for seasonal influenza, with modest annual vaccination rates (<40%) despite favourable attitudes, mirrors international findings among healthcare students and suggests the need to strengthen messages on effectiveness and occupational risk, alongside organisational facilitators (free access, convenient schedules and vaccination points during clinical rotations) [23,27,31]. On the other hand, the observed sex disparity in HPV vaccination (very high coverage in women and clearly lower in men) reflects the history of including only girls in the childhood immunisation schedule and in earlier campaigns [32]. To improve vaccination attitudes among men, previous studies recommend implementing inclusive messaging on HPV vaccination in university settings and specific outreach strategies targeted at male students, in line with current recommendations [21,22,33,34].

Based on the determinants identified in the literature (knowledge, norms, confidence, and structural barriers), it would be advisable to implement a package of measures: institutionalising the review of vaccination status at the beginning of undergraduate studies and revaccination with automated reminders, facilitating influenza vaccination within university premises and during clinical rotations, training clinical communication skills to address vaccine hesitancy, and aligning institutional policies with the high support for mandatory vaccination in high-risk settings [23,27,29,30,31,35]. To achieve this, and given that intention and behaviour are strengthened by regulations and professional role modelling, it is appropriate to involve teaching staff and clinical services as informal prescribers who normalise vaccination throughout the training pathway during undergraduate education [29,30,34].

Overall, the study’s findings emphasize the importance of integrating comprehensive, evidence-based vaccination education into healthcare curricula. Early incorporation of this content in undergraduate programs could help strengthen future professionals’ confidence in vaccines and improve their ability to address hesitancy among patients. Moreover, validated tools such as this questionnaire may facilitate ongoing assessment of vaccination-related knowledge and attitudes, contributing to the design of targeted interventions for both students and healthcare workers. In particular, repeated application of this instrument in educational and clinical settings could support longitudinal monitoring of KAP over time, helping to evaluate the impact of curricular reforms and institutional vaccination policies in healthcare professions [21,22,30,31,36,37]. In addition, its Spanish-language format and focus on vaccines included in national immunisation schedules make it potentially adaptable to other Spanish-speaking countries, where it could inform life-course vaccination strategies and public health planning in the 21st century [21,22,30,31,37].

### Strengths and Limitations

Among the strengths of our study, it is worth highlighting that both the sample size achieved and the response rate were adequate to assess the psychometric properties of this questionnaire in students [35]. All eligible students (n = 483) were invited in a census-like approach, of whom 398 completed the questionnaire (response rate 82.40%). Although no formal a priori power calculation was performed, the final sample clearly exceeds commonly recommended minimums for exploratory factor analysis (e.g., at least 5–10 participants per item and ≥200 participants overall), supporting the adequacy of the sample size for psychometric validation [38,39]. Moreover, the explicit methodological rigour of the validation process, its consistency with national and international validations of other questionnaires, and the convergence observed between the psychometric and descriptive analyses (favourable attitudes but gaps in influenza vaccination and revaccination) reinforce the construct validity and applied usefulness of our questionnaire [21,22,30,31,37]. Furthermore, the sensitivity analysis with Promax rotation recovered the same components as the solution obtained with Varimax rotation with Kaiser normalisation, revealing moderate correlations between them, which is consistent with the interrelated nature of the KAP domains and with the methodological recommendation to use oblique rotations when factors are correlated, thereby improving the interpretability of the results [21,22,30,37].

Another strength of our questionnaire is that it focuses on vaccines included in real immunisation schedules and on routine practices in healthcare settings, which facilitates its use for formative assessment, monitoring of interventions, and decision-making within the university community.

However, several limitations should be considered when interpreting the findings. The cross-sectional design and measurement through a self-administered questionnaire may introduce recall and social desirability biases [40,41]. There may also be some degree of misclassification bias because vaccination status was based on self-report and not on direct verification of vaccination documents or registries, although this was not an objective in this phase of the project. Furthermore, recruiting participants from a single centre, with a predominance of medical students and women, limits generalisability to other degree programmes and universities [42], and the recruitment approach may have favoured participation by more sensitised profiles, with no formal non-response analysis available [30,41,42,43]. The post-pandemic context and the potential influence of local campaigns also add a possible period effect that is difficult to disentangle in a cross-sectional study [30].

From a methodological validation perspective, it should be noted that the evidence of reliability and validity could be strengthened using additional metrics and designs. Temporal stability (test–retest) was not assessed, nor was item performance explored using Rasch models or the possible presence of differential item functioning across subgroups [22,27,31,37]. Finally, although item responses were categorical or ordinal, they were analysed using PCA on Pearson correlations, treating them as approximately continuous in line with Classical Test Theory. This common choice in questionnaire validation may slightly bias factor loadings, so the factorial structure should be interpreted with caution and ideally replicated with methods for ordinal data in future studies [18]. Such analyses would allow a finer calibration of measurement, verification of equivalence between groups, and complementary validation under conditions closer to real practice. As a next step, recent Spanish studies have used Rasch analysis (ordered thresholds, fit statistics, PSI) to complement classical test theory in vaccine knowledge questionnaires, suggesting a useful avenue for refining item functioning and comparability between subgroups [44].

## 5. Conclusions

This study confirmed the reliability and validity of a Spanish-language questionnaire designed to assess knowledge, attitudes, and practices (KAP) regarding vaccination among healthcare students. The final instrument, consisting of 30 items distributed across nine dimensions, showed good internal consistency (Cronbach’s alpha = 0.83) and a coherent factorial structure supported by construct validity analyses. Its multidimensional design enables a nuanced assessment of cognitive, perceptual, and behavioural aspects related to vaccination, including both classical childhood vaccines and newer immunisations.

The descriptive findings indicate that the surveyed students generally exhibited positive attitudes toward vaccination, high confidence in scientific evidence, and good overall compliance with vaccination schedules. However, gaps were identified for specific vaccines and in self-reported awareness of revaccination needs, particularly regarding certain boosters and seasonal influenza. These patterns underline the need to strengthen vaccine-related education, improve information on life-course immunisation, and facilitate access to vaccination during clinical training.

Having a validated instrument such as this questionnaire makes it possible to generate comparable and reliable data on vaccination-related KAP among future healthcare professionals over time. Its application can inform the design and evaluation of targeted educational interventions and institutional strategies aimed at reinforcing trust in vaccines, reducing structural barriers, and promoting evidence-based practice. In addition, its Spanish-language format and alignment with real immunisation schedules support its potential adaptation to other Spanish-speaking settings, contributing to life-course vaccination strategies. Ultimately, fostering adequate knowledge and positive attitudes among healthcare students will strengthen their role as credible agents in immunisation programmes and as key contributors to public health protection in the 21st century.

## Figures and Tables

**Figure 1 vaccines-14-00009-f001:**
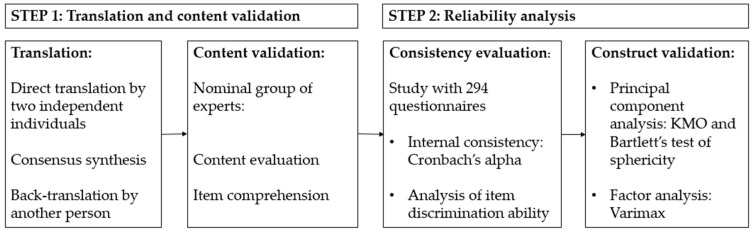
Validation process. Source: authors.

**Table 1 vaccines-14-00009-t001:** Item scale.

Variable	Cronbach’s Alpha	Homogeneity Index
Item 6: Which statement best describes your opinion about vaccines?	0.79	0.01
Item 7: Do you believe that vaccination programs are an effective tool for disease prevention?	0.79	−0.01
Item 8: What most influences your opinion about vaccines?	0.79	0.06
Item 9: Have you been vaccinated according to your country’s immunization schedule?	0.78	0.22
Item 11: Influenza	0.73	0.19
Item 12: Tetanus	0.78	0.38
Item 13: Diphtheria	0.77	0.47
Item 14: Poliomyelitis	0.77	0.53
Item 15: *Haemophilus influenzae* type B	0.76	0.61
Item 16: Pertussis	0.77	0.51
Item 17: Hepatitis A	0.79	0.11
Item 18: Hepatitis B	0.78	0.22
Item 19: Pneumococcus	0.77	0.46
Item 20: Measles	0.77	0.49
Item 21: Mumps	0.77	0.41
Item 22: Rubella	0.77	0.54
Item 23: Meningococcus	0.78	0.40
Item 24: Human papillomavirus (HPV)	0.78	0.16
Item 25: Typhoid fever	0.77	0.52
Item 26: Cholera	0.77	0.55
Item 27: Smallpox	0.77	0.46
Item 28: Tuberculosis	0.78	0.37
Item 29: Respiratory syncytial virus (RSV)	0.77	0.43
Item 30: Varicella (chickenpox)	0.78	0.34
Item 31: Rotavirus	0.77	0.50
Item 32: Herpes zoster	0.78	0.39
Item 33: What other vaccines have you received?	0.79	0.10
Item 34: Are you aware that several vaccines require booster doses for full protection?	0.78	0.21
Item 35: Do you advise your relatives, friends, or colleagues to get vaccinated?	0.78	0.14
Item 36: Do you believe there should be a more specific vaccination program for pregnant women (e.g., against seasonal influenza, mumps, or rubella)?	0.79	0.06
Item 37: What is your opinion about the seasonal influenza vaccine?	0.80	−0.05
Item 38: How often do you get vaccinated against seasonal influenza?	0.79	0.11
Item 39: Do you believe the influenza and hepatitis B vaccines should be mandatory for healthcare professionals (e.g., physicians, nurses)?	0.80	0.04
Item 40: Do you believe the influenza and hepatitis B vaccines should be mandatory for medical students?	0.80	0.01

**Table 2 vaccines-14-00009-t002:** Rotated component matrix.

Variables	Components										
	1	2	3	4	5	6	7	8	9	10	11
Item 26: Cholera	0.80										
Item 25: Typhoid fever	0.78										
Item 27: Smallpox	0.68										
Item 28: Tuberculosis	0.57										
Item 29: Respiratory syncytial virus	0.54										
Item 31: Rotavirus	0.51										
Item 14: Poliomyelitis		0.77									
Item 13: Diphtheria		0.75									
Item 15: *Haemophilus influenzae* t. B		0.61									
Item 16: Pertussis		0.60									
Item 12: Tetanus		0.46									
Item 20: Measles			0.64								
Item 22: Rubella		0.46	0.62								
Item 19: Pneumococcus			0.61								
Item 21: Mumps			0.57								
Item 6: Which statement best describes your opinion about vaccines?				0.96							
Item 7: Do you believe that vaccination programs are an effective tool for disease prevention?				0.96							
Item 36 Do you believe there should be a more specific vaccination program for pregnant women (for example, against seasonal influenza, mumps, or rubella)?				0.49							
Item 39: Do you believe the influenza and hepatitis B vaccines should be mandatory for healthcare professionals?					0.94						
Item 40: Do you believe the influenza and hepatitis B vaccines should be mandatory for medical students?					0.93						
Item 18: Hepatitis B						0.87					
Item 17: Hepatitis A						0.83					
Item 38: How often do you get vaccinated against seasonal influenza?							0.75				
Item 11: Influenza							0.74				
Item 9: Have you been vaccinated according to your country’s immunization schedule?								0.66			
Item 30: Chickenpox								0.53			
Item 34: Are you aware that several vaccines require booster doses for full protection?											
Item 24: Human papillomavirus (HPV)									0.73		
Item 23: Meningococcus			0.46						0.49		
Item 37: What is your opinion about the seasonal influenza vaccine?									−0.48		
Item 35: Do you advise your relatives, friends, or colleagues to get vaccinated?										0.76	
Item 32: Herpes zoster										−0.48	
Item 33: What other vaccines have you received?											0.74
Item 8: What most influences your opinion about vaccines?										0.46	0.47

Extraction method: Principal Component Analysis. Rotation method: Varimax normalization with Kaiser. Rotation converged in 15 iterations.

**Table 3 vaccines-14-00009-t003:** Recalculated item scale.

Variable	Cronbach’s Alpha	Homogeneity Index
Item 8: What most influences your opinion about vaccines?	0.82	0.15
Item 9: Have you been vaccinated according to your country’s immunization schedule?	0.82	0.28
Item 11: Influenza	0.79	0.24
Item 12: Tetanus	0.82	0.41
Item 13: Diphtheria	0.82	0.48
Item 14: Poliomyelitis	0.81	0.52
Item 15: *Haemophilus influenzae* type B	0.81	0.59
Item 16: Pertussis	0.82	0.47
Item 17: Hepatitis A	0.83	0.16
Item 18: Hepatitis B	0.82	0.28
Item 19: Pneumococcus	0.81	0.45
Item 20: Measles	0.81	0.49
Item 21: Mumps	0.81	0.42
Item 22: Rubella	0.81	0.52
Item 23: Meningococcus	0.82	0.43
Item 24: Human papillomavirus (HPV)	0.82	0.20
Item 25: Typhoid fever	0.81	0.51
Item 26: Cholera	0.81	0.53
Item 27: Smallpox	0.81	0.46
Item 28: Tuberculosis	0.82	0.35
Item 29: Respiratory syncytial virus (RSV)	0.81	0.42
Item 30: Varicella (chickenpox)	0.82	0.37
Item 31: Rotavirus	0.81	0.47
Item 32: Herpes zoster	0.82	0.41
Item 33: What other vaccines have you received?	0.83	0.15
Item 34: Are you aware that several vaccines require booster doses for full protection?	0.82	0.25
Item 35: Do you advise your relatives, friends, or colleagues to get vaccinated?	0.82	0.18
Item 38: How often do you get vaccinated against seasonal influenza?	0.82	0.17
Item 39: Do you believe the influenza and hepatitis B vaccines should be mandatory for healthcare professionals (e.g., physicians, nurses)?	0.83	0.09
Item 40: Do you believe the influenza and hepatitis B vaccines should be mandatory for medical students?	0.83	0.07

**Table 4 vaccines-14-00009-t004:** Recalculated component matrix.

Variables	Components								
	1	2	3	4	5	6	7	8	9
Item 14: Poliomyelitis	0.79								
Item 13: Diphtheria	0.78								
Item 15: *Haemophilus influenzae* t. B	0.67								
Item 16: Pertussis	0.66								
Item 22: Rubeola	0.59		0.44						
Item 12: Tetanus	0.46					0.46			
Item 26: Cholera		0.83							
Item 25: Typhoid fever		0.82							
Item 27: Smallpox		0.66							
Item 29: Respiratory syncytial virus (RSV)		0.57							
Item 28: Tuberculosis		0.57	0.45						
Item 31: Rotavirus		0.47	0.43						
Item 23: Meningococcus			0.59						
Item 19: Pneumococcus			0.55						
Item 21: Mumps	0.42		0.43						
Item 40: Do you believe the influenza and hepatitis B vaccines should be mandatory for medical students?				0.94					
Item 39: Do you believe the influenza and hepatitis B vaccines should be mandatory for healthcare professionals?				0.94					
Item 18: Hepatitis B					0.88				
Item 17: Hepatitis A					0.83				
Item 30: Chickenpox						0.67			
Item 20: Measles			0.48			0.51			
Item 9: Have you been vaccinated according to your country’s immunization schedule?						0.43			
Item 38: How often do you get vaccinated against seasonal influenza?							0.74		
Item 11: Influenza							0.71		
Item 34: Are you aware that several vaccines require booster doses for full protection?							0.41		
Item 33: What other vaccines have you received?								0.68	
Item 24: Human papillomavirus (HPV)								−0.54	
Item 8: What most influences your opinion about vaccines?									0.73
Item 35: Do you advise your relatives, friends, or colleagues to get vaccinated?									0.59
Item 32: Herpes zoster		0.43							−0.45

Extraction method: Principal Component Analysis. Rotation method: Varimax normalization with Kaiser. Rotation converged in 13 iterations.

**Table 5 vaccines-14-00009-t005:** General characteristics of participants.

Variable	Category	n (%)
Gender	Female	308 (77.5)
	Male	78 (19.6)
	Other	2 (0.5)
Country of study	Spain	373 (93.7)
	Other	25 (6.3)
Degree program	Medicine	306 (77.0)
	Nursing	60 (15.2)
	Other health-related degrees	32 (8.0)
Academic year	Fifth year	242 (60.8)
	Other years	156 (39.2)
Mean age (years)		23.78 ± 3.77 (range: 18–56)

**Table 6 vaccines-14-00009-t006:** Childhood vaccines.

Item	Total	Gender		Studies		Academic Year	
Answer	(n = 398)	Men	Women	Medicine	Other	5th	Other
		(n = 80)	(n = 316)	(n = 314)	(n = 84)	(n = 248)	(n = 150)
Item 12. Tetanus							
A. Yes	363 (91.70)	72 (90.00)	291 (92.10)	299 (95.20)	66 (78.60)	236 (95.20)	129 (78.60)
B. No	6 (1.50)	0 (0.00)	6 (1.90)	2 (0.60)	4 (4.80)	2 (0.80)	4 (4.80)
C. I don’t know	22 (5.60)	7 (8.80)	15 (4.70)	10 (3.20)	12 (14.30)	8 (3.20)	14 (14.30)
		*p* = 0.18		*p* < 0.01		*p* < 0.01	
Item 13. Diphtheria							
A. Yes	297 (75.00)	57 (71.30)	240 (75.90)	257 (81.80)	42 (50.00)	208 (8.90)	91 (60.70)
B. No	8 (2.00)	4 (5.00)	4 (1.30)	4 (1.30)	4 (4.80)	2 (0.80)	6 (4.000)
C. I don’t know	75 (18.90)	15 (18.80)	60 (19.00)	46 (14.60)	29 (34.50)	33 (13.30)	42 (28.0)
		*p* = 0.10		*p* < 0.01		*p* < 0.01	
Item 14. Poliomyelitis							
A. Yes	274 (69.20)	56 (70.00)	218 (69.0)	240 (76.40)	36 (42.90)	195 (78.60)	81 (54.00)
B. No	12 (3.00)	2 (2.50)	10 (3.2)	10 (3.20)	2 (2.40)	9 (3.60)	3 (2.00)
C. I don’t know	87 (22.00)	16 (20.00)	71 (22.50)	52 (16.60)	35 (41.70)	34 (13.70)	53 (35.30)
		*p* = 0.88		*p* < 0.01		*p* < 0.01	
Item 15. *Haemophilus Influenzae* type B							
A. Yes	273 (68.90)	51 (63.80)	222 (70.30)	235 (74.80)	39 (46.40)	190 (76.60)	84 (56.00)
B. No	17 (4.30)	6 (7.50)	11 (3.50)	16 (5.10)	2 (2.40)	14 (5.60)	4 (2.70)
C. I don’t know	86 (21.70)	16 (20.00)	70 (22.20)	53 (16.90)	33 (39.30)	37 (14.90)	49 (32.70)
		*p* = 0.24		*p* < 0.01		*p* < 0.01	
Item 16. Pertussis							
A. Yes	171 (43.20)	39 (48.80)	132 (41.80)	159 (50.60)	14 (16.70)	124 (50.00)	49 (32.70)
B. No	33 (8.30)	6 (7.50)	27 (8.50)	27 (8.60)	6 (7.10)	23 (9.30)	10 (6.70)
C. I don’t know	155 (39.10)	29 (36.30)	126 (39.90)	108 (34.40)	47 (56.00)	86 (34.70)	69 (46.00)
		*p* = 0.62		*p* < 0.01		*p* < 0.01	
Item 22. Rubella							
A. Yes	65 (87.20)	65 (85.50)	270 (87.70)	277 (88.20)	60 (71.40)	217 (87.50)	120 (80.00)
B. No	2 (2.10)	2 (2.60)	6 (1.90)	5 (1.60)	30 (35.70)	3 (1.20)	5 (3.30)
C. I don’t know	9 (10.70)	9 (11.80)	32 (10.40)	22 (7.00)	19 (22.60)	18 (7.30)	23 (15.30)
		*p* = 0.09		*p* < 0.01		*p* < 0.01	

The results are presented as n (%). *p* from Fisher’s exact test.

**Table 7 vaccines-14-00009-t007:** Special or less frequent vaccines *.

Item	Total	Gender		Studies		Academic Year	
Answer	(n = 398)	Men	Women	Medicine	Other	5th	Other
		(n = 80)	(n = 316)	(n = 314)	(n = 84)	(n = 248)	(n = 150)
Item 25. Typhoid fever							
A. Yes	68 (16.70)	16 (20.00)	50 (15.80)	56 (17.80)	12 (14.30)	40 (16.10)	28 (18.70)
B. No	116 (29.30)	24 (30.00)	92 (29.10)	106 (33.80)	10 (11.90)	90 (36.30)	26 (17.30)
C. I don’t know	181 (45.70)	35 (43.80)	146 (46.20)	135 (43.00)	46 (54.80)	104 (41.90)	77 (51.30)
		*p* = 0.70		*p* < 0.01		*p* < 0.01	
Item 26. Cholera							
A. Yes	52 (13.10)	13 (16.30)	39 (12.30)	41 (13.10)	12 (14.30)	26 (10.50)	27 (18.00)
B. No	132 (33.30)	26 (32.50)	106 (33.50)	124 (39.50)	9 (10.70)	109 (44.00)	24 (16.00)
C. I don’t know	178 (44.90)	36 (45.00)	142 (44.90)	130 (41.40)	48 (57.10)	98 (39.50)	80 (53.30)
		*p* = 0.71		*p* < 0.01		*p* < 0.01	
Item 27. Smallpox							
A. Yes	120 (29.80)	27 (33.80)	91 (28.80)	97 (30.90)	23 (27.40)	67 (27.00)	53 (35.30)
B. No	109 (27.50)	18 (22.50)	91 (28.80)	101 (32.20)	8 (9.50)	90 (36.30)	19 (12.70)
C. I don’t know	142 (35.90)	30 (37.50)	112 (35.40)	100 (31.80)	42 (50.00)	78 (31.50)	64 (42.70)
		*p* = 0.47		*p* < 0.01		*p* < 0.01	
Item 28. Tuberculosis							
A. Yes	89 (22.50)	15 (18.80)	74 (23.40)	58 (18.50)	31 (36.90)	37 (14.90)	52 (34.70)
B. No	161 (40.70)	39 (48.80)	122 (38.60)	151 (48.10)	11 (13.10)	131 (52.80)	31 (20.70)
C. I don’t know	119 (30.10)	20 (25.00)	99 (31.30)	88 (28.00)	31 (36.90)	65 (26.20)	54 (36.00)
		*p* = 0.21		*p* < 0.01		*p* < 0.01	
Item 29. Respiratory syncytial virus (RSV)							
A. Yes	92 (23.20)	18 (22.50)	74 (23.40)	72 (22.90)	21 (25.00)	51 (20.60)	42 (28.00)
B. No	132 (33.30)	32 (40.00)	100 (31.60)	124 (39.50)	9 (10.70)	105 (42.30)	28 (18.70)
C. I don’t know	140 (35.40)	25 (31.30)	115 (36.40)	102 (32.50)	38 (45.20)	79 (31.90)	61 (40.70)
		*p* = 0.41		*p* < 0.01		*p* < 0.01	
Item 31. Rotavirus							
A. Yes	128 (32.30)	25 (31.30)	103 (32.60)	94 (29.90)	36 (42.90)	70 (28.20)	60 (40.00)
B. No	80 (20.20)	21 (26.30)	59 (18.70)	73 (23.20)	7 (8.30)	64 (25.80)	16 (10.70)
C. I don’t know	160 (40.40)	29 (36.30)	131 (41.50)	139 (44.30)	31 (36.90)	100 (40.30)	60 (40.00)
		*p* < 0.01		*p* < 0.01		*p* < 0.01	

The results are presented as n (%). *p* from Fisher’s exact test. * These items do not correspond to universal calendar vaccines, but to vaccines indicated only for particular risk groups or contexts.

**Table 8 vaccines-14-00009-t008:** Respiratory vaccines.

Item	Total	Gender		Studies		Academic Year	
Answer	(n = 398)	Men	Women	Medicine	Other	5th	Other
		(n = 80)	(n = 316)	(n = 314)	(n = 84)	(n = 248)	(n = 150)
Item 19. Pneumococcus							
A. Yes	272 (68.70)	47 (58.80)	225 (71.20)	221 (70.40)	52 (61.90)	173 (69.80)	100 (66.70)
B. No	27 (6.80)	8 (10.00)	19 (6.00)	26 (8.30)	2 (2.40)	21 (8.50)	7 (4.70)
C. I don’t know	80 (20.20)	20 (25.00)	60 (19.00)	60 (19.10)	20 (23.80)	48 (19.40)	32 (21.30)
		*p* = 0.13		*p* = 0.12		*p* = 0.36	
Item 20. Measles							
A. Yes	362 (90.90)	67 (83.80)	293 (92.70)	295 (93.90)	67 (79.80)	232 (93.50)	130 (86.70)
B. No	7 (1.80)	3 (3.80)	4 (1.30)	3 (1.00)	4 (4.80)	2 (0.80)	5 (3.30)
C. I don’t know	22 (5.60)	7 (8.80)	15 (4.70)	12 (3.80)	10 (11.90)	10 (4.00)	12 (8.00)
		*p* = 0.10		*p* < 0.01		*p* < 0.05	
Item 21. Mumps							
A. Yes	302 (76.30)	62 (77.50)	240 (75.90)	244 (77.70)	60 (71.40)	189 (76.20)	115 (76.70)
B. No	14 (3.50)	4 (5.00)	10 (3.20)	12 (3.80)	2 (2.40)	10 (4.00)	4 (2.70)
C. I don’t know	80 (17.20)	10 (12.50)	58 (18.40)	50 (15.90)	18 (21.40)	42 (16.90)	26 (17.30)
		*p* = 0.40		*p* = 0.39		*p* = 0.78	
Item 23. Meningococcus							
A. Yes	331 (83.60)	59 (73.80)	272 (86.10)	266 (84.70)	67 (79.80)	212 (85.50)	121 (80.70)
B. No	13 (3.30)	4 (5.00)	9 (2.80)	11 (3.50)	2 (2.40)	5 (2.00)	8 (5.30)
C. I don’t know	36 (9.10)	11 (13.80)	25 (7.90)	26 (8.30)	10 (11.90)	20 (8.10)	16 (10.70)
		*p* = 0.11		*p* = 0.50		*p* = 0.13	

The results are presented as n (%). *p* from Fisher’s exact test.

**Table 9 vaccines-14-00009-t009:** Hepatitis.

Item	Total	Gender		Studies		Academic Year	
Answer	(n = 398)	Men	Women	Medicine	Other	5th	Other
		(n = 80)	(n = 316)	(n = 314)	(n = 84)	(n = 248)	(n = 150)
Item 17. Hepatitis A							
A. Yes	230 (58.10)	46 (57.50)	184 (58.20)	171 (54.50)	60 (71.4)	125 (50.40)	106 (70.70)
B. No	85 (21.50)	19 (23.80)	66 (20.90)	71 (22.60)	15 (17.9)	61 (24.60)	25 (16.70)
C. I don’t know	63 (15.90)	12 (15.00)	51 (16.10)	60 (19.10)	3 (3.6)	52 (21.00)	11 (7.30)
		*p* = 0.86		*p* < 0.01		*p* < 0.01	
Item 18. Hepatitis B							
A. Yes	341 (86.10)	68 (85.00)	273 (86.40)	268 (85.4)	75 (89.30)	207 (83.50)	136 (90.70)
B. No	6 (1.50)	1 (1.30)	5 (1.60)	4 (1.3)	4 (2.40)	4 (1.60)	2 (1.30)
C. I don’t know	37 (9.30)	10 (12.50)	27 (8.50)	33 (10.5)	33 (4.80)	30 (12.10)	7 (4.70)
		*p* = 0.58		*p* = 0.22		*p* < 0.05	

The results are presented as n (%). *p* from Fisher’s exact test.

**Table 10 vaccines-14-00009-t010:** Attitudes.

Item	Total	Gender		Studies		Academic Year	
Answer	(n = 398)	Men	Women	Medicine	Other	5th	Other
		(n = 80)	(n = 316)	(n = 314)	(n = 84)	(n = 248)	(n = 150)
Item 8. What most influences your opinion about vaccines?							
A. Scientific data	296 (74.70)	56 (70.00)	239 (75.60)	238 (75.80)	58 (69.00)	189 (76.20)	107 (71.30)
B. Social media	4 (1.00)	2 (2.50)	2 (0.60)	3 (1.00)	1 (1.20)	2 (0.80)	2 (1.30)
C. Experienced physicians; professors	93 (23.50)	22 (27.50)	70 (22.20)	70 (22.30)	23 (27.40)	55 (22.20)	28 (18.70)
D. My family members	3 (0.80)	0 (0.00)	3 (0.90)	2 (0.60)	1 (1.20)	1(0.40)	2 (1.30)
E. Religious beliefs	0 (0.00)	0 (0.00)	0 (0.00)	0 (0.00)	0 (0.00)	0 (0.00)	0 (0.00)
F. My friends; colleagues	0 (0.00)	0 (0.00)	0 (0.00)	0 (0.00)	0 (0.00)	0 (0.00)	0 (0.00)
		*p* = 0.26		*p* = 0.69		*p* = 0.57	
Item 24. Human papillomavirus (HPV)							
A. Yes	320 (80.40)	32 (40.0)	286 (90.5)	262 (83.4)	58 (69.0)	213 (85.9)	107 (71.3)
B. No	45 (11.30)	30 (37.5)	15 (4.7)	34 (10.8)	11 (13.1)	22 (8.9)	23 (15.3)
C. I don’t know	18 (4.50)	12 (15.0)	6 (1.9)	11 (3.5)	7 (8.3)	6 (2.4)	12 (8.0)
		*p* < 0.01		*p* = 0.07		*p* < 0.01	
Item 35. Do you advise your relatives, friends, or colleagues to get vaccinated?							
A. Yes	381 (95.70)	77 (96.30)	302 (77.80)	304 (96.80)	77 (91.70)	240 (96.80)	141 (94.00)
B. No	3 (0.80)	0 (0.00)	3 (0.80)	3 (1.00)	0 (0.00)	1 (0.40)	2 (1.30)
C. I don’t know	4 (1.00)	1 (1.30)	3 (0.80)	1 (0.30)	3 (3.60)	1 (0.40)	3 (2.00)
		*p* = 0.66		*p* = 0.02		*p* = 0.17	

The results are presented as n (%). *p* from Fisher’s exact test.

**Table 11 vaccines-14-00009-t011:** Mandates.

Item	Total	Gender		Studies		Academic Year	
Answer	(n = 398)	Men	Women	Medicine	Other	5th	Other
		(n = 80)	(n = 316)	(n = 314)	(n = 84)	(n = 248)	(n = 150)
Item 39: Do you believe the influenza and hepatitis B vaccines should be mandatory for healthcare professionals?							
A. No, because everyone should have the right to choose.	92 (23.10)	21 (26.30)	71 (22.50)	73 (23.20)	19 (22.60)	64 (25.80)	28 (18.70)
B. No, because those vaccines are not effective.	0 (0.00)	0 (0.00)	0 (0.00)	0 (0.00)	0 (0.00)	0 (0.00)	0 (0.00)
C. No, because those vaccines are not safe.	1 (0.30)	0 (0.00)	0 (0.00)	1 (0.30)	0 (0.00)	1 (0.40)	0 (0.00)
D. Yes, because healthcare personnel are more likely to become infected and spread the virus.	303 (76.10)	58 (72.50)	244 (77.20)	239 (76.10)	64 (76.20)	182 (73.40)	121 (80.70)
		*p* = 0.45		*p* = 0.87		*p* = 0.19	
Item 40. Do you believe the influenza and hepatitis B vaccines should be mandatory for medical students?							
A. No, because everyone should have the right to choose.	112 (28.10)	27 (33.80)	84 (26.60)	93 (29.60)	19 (22.60)	83 (33.50)	29 (19.30)
B. No, because those vaccines are not effective.	1 (0.30)	0 (0.00)	1 (0.30)	1 (0.30)	0 (0.00)	1 (0.40)	0 (0.00)
C. No, because those vaccines are not safe.	0 (0.00)	0 (0.00)	0 (0.00)	0 (0.00)	0 (0.00)	0 (0.00)	0 (0.00)
D. Yes, because healthcare personnel are more likely to become infected and spread the virus.	284 (71.40)	53 (66.30)	230 (72.80)	220 (70.10)	64 (76.20)	164 (66.10)	120 (80.00)
		*p* = 0.41		*p* = 0.41		*p* < 0.01	

The results are presented as n (%). *p* from Fisher’s exact test.

**Table 12 vaccines-14-00009-t012:** Schedule.

Item	Total	Gender		Studies		Academic Year	
Answer	(n = 398)	Men	Women	Medicine	Other	5th	Other
		(n = 80)	(n = 316)	(n = 314)	(n = 84)	(n = 248)	(n = 150)
Item 9. Have you been vaccinated according to your country’s immunization schedule?							
A. Yes	393 (99.20)	77 (96.30)	314 (99.40)	309 (98.40)	84 (100.00)	243 (98.00)	150 (100.00)
B. No	0 (0.00)	0 (0.00)	0 (0.00)	0 (0.00)	0 (0.00)	0 (0.00)	0 (0.000)
C. I don’t know	3 (0.80)	2 (2.50)	1 (0.30)	3 (1.00)	0 (0.00)	3 (1.20)	0 (0.0)
		*p* = 0.04		*p* = 0.37		*p* = 0.18	
Item 20. Measles							
A. Yes	362 (90.90)	67 (83.80)	293 (92.70)	295 (93.90)	67 (79.80)	232 (93.50)	130 (86.70)
B. No	7 (1.80)	3 (3.80)	4 (1.30)	3 (1.00)	4 (4.80)	2 (0.80)	5 (3.30)
C. I don’t know	22 (5.60)	7 (8.80)	15 (4.70)	12 (3.80)	10 (11.90)	10 (4.00)	12 (8.00)
		*p* = 0.10		*p* < 0.01		*p* < 0.05	
Item 30. Varicella (Chickepox)							
A. Yes	247 (62.10)	44 (55.00)	202 (63.90)	200 (63.70)	47 (56.00)	161 (64.90)	86 (57.30)
B. No	119 (29.80)	25 (31.30)	93 (29.40)	90 (28.70)	29 (34.50)	69 (27.80)	50 (33.30)
C. I don’t know	22 (5.60)	8 (10.00)	14 (4.40)	29 (9.20)	5 (6.00)	13 (5.20)	9 (6.00)
		*p* = 0.11		*p* = 0.49		*p* = 0.39	

The results are presented as n (%). *p* from Fisher’s exact test.

**Table 13 vaccines-14-00009-t013:** Influenza.

Item	Total	Gender		Studies		Academic Year	
Answer	(n = 398)	Men	Women	Medicine	Other	5th	Other
		(n = 80)	(n = 316)	(n = 314)	(n = 84)	(n = 248)	(n = 150)
Item 11. Influenza							
A. Yes	227 (57.30)	45 (56.30)	182 (57.60)	173 (55.10)	54 (64.30)	136 (54.8)	91 (60.7)
B. No	139 (35.10)	26 (32.50)	113 (35.80)	120 (38.20)	21 (25.00)	94 (37.9)	47 (31.3)
C. I don’t know	18 (4.50)	6 (7.50)	12 (3.80)	14 (4.50)	4 (4.80)	12 (4.8)	6 (4)
		*p* = 0.34		*p* = 0.12		*p* = 0.402	
Item 34. Are you aware that several vaccines require booster doses for full protection?							
A. Yes, I am aware of it and I do it correctly.	292 (73.40)	57 (71.30)	233 (73.70)	224 (71.30)	68 (81.00)	171 (69.00)	121 (80.70)
B. Yes, I am aware of it, but I am not sure if my vaccination is complete.	105 (26.40)	23 (28.80)	82 (25.90)	89 (28.30)	16 (19.00)	76 (30.60)	29 (19.30)
C. No, this is the first time I’ve heard about it.	1 (0.30)	0 (0.00)	1 (0.30)	1 (0.30)	0 (0.00)	1 (0.40)	0 (0.00)
D. No, it’s not necessary because vaccination always provides lifelong protection.	0 (0.00)	0 (0.00)	0 (0.00)	0 (0.00)	0 (0.00)	0 (0.00)	0 (0.00)
		*p* = 0.78		*p* = 0.20		*p* < 0.05	
Item 38. How often do you get vaccinated against seasonal influenza?							
A. Every two seasons.	18 (4.50)	4 (5.00)	14 (4.40)	13 (4.10)	5 (6.00)	11 (2.80)	7 (4.70)
B. Every season.	159 (39.90)	35 (43.80)	124 (39.20)	133 (42.40)	26 (31.00)	106 (26.90)	53 (35.30)
C. I have never been vaccinated against seasonal flu.	73 (18.30)	14 (17.50)	58 (18.40)	57 (18.20)	16 (19.00)	45 (11.40)	28 (18.70)
D. I have been vaccinated only once.	78 (19.60)	11 (13.80)	66 (20.90)	61 (15.50)	17 (20.20)	46 (11.70)	32 (21.30)
E. Not regularly.	66 (16.60)	15 (18.80)	51 (16.10)	46 (14.60)	20 (23.80)	37 (9.40)	29 (19.30)
		*p* = 0.68		*p* = 0.20		*p* = 0.60	

The results are presented as n (%). *p* from Fisher’s exact test.

**Table 14 vaccines-14-00009-t014:** New vaccines.

Item	Total	Gender		Studies		Academic Year	
Answer	(n = 398)	Men	Women	Medicine	Other	5th	Other
		(n = 80)	(n = 316)	(n = 314)	(n = 84)	(n = 248)	(n = 150)
P32. Herpes Zoster							
A. Yes	82 (20.50)	15 (18.80)	66 (20.90)	50 (15.90)	32 (38.10)	32 (12.90)	50 (33.30)
B. No	176 (44.20)	42 (52.50)	133 (42.10)	166 (52.90)	10 (11.90)	141 (56.90)	35 (23.30)
C. I don’t know	111 (28.00)	18 (22.50)	93 (29.40)	81 (25.80)	30 (35.70)	61 (24.60)	50 (33.30)
		*p* = 0.25		*p* < 0.01		*p* < 0.01	

The results are presented as n(%). *p* from Fisher’s exact test.

## Data Availability

Data availability is under petition.

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
