# Peer review of "Vaccination Knowledge, Attitudes and Practices Among Healthcare Students in Spain: Development and Psychometric Validation of a Life-Course Immunization Questionnaire"

_vaccines, 2025, doi:10.3390/vaccines14010009_

Round 1
Reviewer 1 Report
Comments and Suggestions for Authors
vaccine hesitancy has been recognized as one of the main health threats of
the 21st century, as it weakens the success of vaccination programs .
In the general population, vaccine acceptance depends on multiple
factors, and these challenges have become particularly evident in the 21st century, in the wake of the COVID-19 pandemic, the rapid development of novel vaccine
platforms, and the amplification of misinformation through digital and social media. The study evaluate the reliability and validity of a Spanish-language questionnaire
designed to assess knowledge, attitudes, and practices (KAP) regarding vaccination
among healthcare students.
The final instrument, consisting of 30 items distributed across nine dimensions, showed good internal consistency (Cronbach’s alpha = 0.83) and a co- herent factorial structure supported by construct validity analyses. Its multidimensional design enables a nuanced assessment of cognitive, perceptual, and behavioural aspects . Having a validated instrument such as this questionnaire makes it possible to generate comparable and reliable data on vaccination-related KAP among future healthcare professionals over time.
I think that the paper is well written and and may be very useful .
I think may be acceptes in the present form
Author Response
Vaccine hesitancy has been recognized as one of the main health threats of the 21st century, as it weakens the success of vaccination programs .
In the general population, vaccine acceptance depends on multiple factors, and these challenges have become particularly evident in the 21st century, in the wake of the COVID-19 pandemic, the rapid development of novel vaccine platforms, and the amplification of misinformation through digital and social media. The study evaluate the reliability and validity of a Spanish-language questionnaire designed to assess knowledge, attitudes, and practices (KAP) regarding vaccination among healthcare students.
The final instrument, consisting of 30 items distributed across nine dimensions, showed good internal consistency (Cronbach’s alpha = 0.83) and a co- herent factorial structure supported by construct validity analyses. Its multidimensional design enables a nuanced assessment of cognitive, perceptual, and behavioural aspects . Having a validated instrument such as this questionnaire makes it possible to generate comparable and reliable data on vaccination-related KAP among future healthcare professionals over time.
I think that the paper is well written and and may be very useful .
I think may be acceptes in the present form.
We sincerely thank Reviewer for the careful reading of our manuscript and for their very positive assessment.
We appreciate the reviewer’s summary of our work, particularly the recognition that: (1) vaccine hesitancy is a key health challenge in the 21st century and an important context for our study; (2) the validated Spanish-language KAP questionnaire shows good internal consistency and a coherent factorial structure;(3) the multidimensional design of the instrument allows a nuanced assessment of cognitive, perceptual, and behavioural aspects related to vaccination; (4) the tool may be useful for generating comparable and reliable data on vaccination-related KAP among future healthcare professionals over time.
As the reviewer kindly indicates that the manuscript may be accepted in its present form and does not request specific modifications, we have not introduced substantive changes to the study design or results based on this report. We will, however, revise the manuscript according to the suggestions made by the other reviewers, and any changes introduced in the new version will be those recommended in response to their comments. In addition, we have carefully re-read the manuscript and made minor language and formatting improvements to enhance clarity and readability, without altering the content or conclusions.
We are grateful for the reviewer’s supportive comments and the consideration of our work for publication in Vaccines.
Reviewer 2 Report
Comments and Suggestions for Authors
The results of this manuscript are confirm that the questionnaire has a solid, reliable, and coherent factorial structure that allows for differentiated assessment of knowledge, attitudes, and practices related to vaccination among healthcare students. The main innovation of the questionnaire was the inclusion of self-reported vaccination status, allowing comparisons between knowledge, attitudes, and real practices.
The study’s findings emphasize the importance of integrating comprehensive, evidence-based vaccination education into healthcare curricula. Validated tools such as this questionnaire may facilitate ongoing assessment of vaccination-related knowledge and attitudes, contributing to the design of targeted interventions for both students and healthcare workers. Its Spanish-language format and focus on vaccines included in national immunization schedules make it potentially adaptable to other Spanish-speaking countries.
However, my proposition for authors:
- Exclude the vaccines which not contain in vaccination calendar from the questionnaire. For example: Typhoid fever, Cholera, Respiratory syncytial virus etc. Besides the questionnaire not based on the vaccinations' documents.
- What it is mean? :"Table 60. 90% of the total variance." - line 212. Perhaps -it is mistake.
Author Response
The results of this manuscript are confirm that the questionnaire has a solid, reliable, and coherent factorial structure that allows for differentiated assessment of knowledge, attitudes, and practices related to vaccination among healthcare students. The main innovation of the questionnaire was the inclusion of self-reported vaccination status, allowing comparisons between knowledge, attitudes, and real practices.
The study’s findings emphasize the importance of integrating comprehensive, evidence-based vaccination education into healthcare curricula. Validated tools such as this questionnaire may facilitate ongoing assessment of vaccination-related knowledge and attitudes, contributing to the design of targeted interventions for both students and healthcare workers. Its Spanish-language format and focus on vaccines included in national immunization schedules make it potentially adaptable to other Spanish-speaking countries.
We sincerely thank Reviewer for the careful evaluation of our manuscript and for their positive comments regarding the solidity, reliability, and usefulness of the validated questionnaire, as well as its potential applicability in other Spanish-speaking settings.
However, my proposition for authors:
1.- Exclude the vaccines which not contain in vaccination calendar from the questionnaire. For example: Typhoid fever, Cholera, Respiratory syncytial virus etc. Besides the questionnaire not based on the vaccinations' documents.
We appreciate this thoughtful comment and the opportunity to clarify the scope and rationale of our instrument.
Our questionnaire was explicitly designed as a life-course immunisation tool aimed at healthcare students, and therefore: (1) It includes vaccines from the Spanish national immunisation schedule, (2) As well as selected vaccines recommended for specific risk groups or situations (e.g., travel-related vaccines such as typhoid or cholera, or vaccines indicated for particular clinical or occupational contexts).
These “special or less frequent vaccines” are grouped and reported as such (in Table 7), and were included because they are relevant in the training and future professional practice of healthcare students, who will often have to advise patients about them and, in some cases, receive them themselves.
In light of your suggestion, in the revised version we have:
1.- Clarified in the Materials and Methods section (questionnaire description) that the instrument includes both:
1.a.- vaccines from the routine national immunisation schedule, and
1.b.- certain vaccines recommended in specific epidemiological, travel, or occupational circumstances.
2.- Explicitly specified in the caption/description of the table on “special or less frequent vaccines” that these items do not correspond to universal calendar vaccines, but to vaccines indicated only for particular risk groups or contexts.
We have chosen not to exclude these vaccines from the questionnaire at this stage because they form part of the validated factor structure and contribute to the conceptual aim of capturing life-course vaccination within healthcare settings. However, we have now made this scope more explicit in the text and we acknowledge that, for future adaptations to other countries, the module on special vaccines could be tailored to the local immunisation schedules and recommendations.
Regarding the reviewer’s remark that “the questionnaire not based on the vaccinations' documents”, we understand that this may refer to the fact that vaccination status was self-reported rather than verified against official vaccination records. In the manuscript we already state that data on vaccination were self-reported, and in the Strengths and limitations section we explicitly acknowledge the possibility of misclassification due to not cross-checking with vaccination records. To improve clarity, in the revised version we have slightly refined this wording to state that: “There may also be some degree of misclassification bias because vaccination status was based on self-report and not on direct verification of vaccination documents or registries, although this was not an objective in this phase of the project.”
2.- What it is mean? :"Table 60. 90% of the total variance." - line 212. Perhaps -it is mistake.
We thank the reviewer for spotting this issue. We confirm that this was a typographical error and an artefact from an earlier draft of the PCA description. It does not correspond to any table in the final version of the manuscript and could indeed be confusing.
In the revised manuscript, we have removed this erroneous sentence (“Table 60. 90% of the total variance…”) so that the Results section now refers only to the correct tables and the final factor solution (nine components explaining approximately 61% of the variance).
Once again, we thank Reviewer for their constructive feedback, which has helped us to improve the clarity and precision of the manuscript.
Reviewer 3 Report
Comments and Suggestions for Authors
Lines 52-55. Provide a citation.
Methods
Line 240. A total of 483 individuals were invited to participate, of whom 398 completed the survey, yielding a response rate of 82.40%
Comment: How was the sample size determined?
Line 195. Table. The table variables were narrated in the methodology?
Author Response
1.- Lines 52-55. Provide a citation.
We thank the reviewer for this observation. We agree that the statement in lines 52–55, where we refer to vaccine hesitancy in the context of the COVID-19 pandemic, novel vaccine platforms, and the amplification of misinformation through digital and social media, should be explicitly supported by references.
In the revised manuscript, we have added citations to:
WHO documents and summaries that identify vaccine hesitancy as one of the top ten threats to global health in 2019, and highlight its impact on immunization programmes in the 21st century.
Reviews on social media and vaccine hesitancy during COVID-19, which describe how misinformation and disinformation on digital platforms have affected vaccine confidence and uptake (Puri et al., 2020; Cascini et al., 2022).
Empirical evidence on the impact of COVID-19 vaccine misinformation on vaccination intentions, specifically the study by Loomba et al. (2021), which quantifies how exposure to misinformation reduces vaccine intent in the UK and USA.
In the Introduction, the sentence now reads (revised for clarity and with citations added):“These challenges have become particularly evident in the 21st century, in the wake of the COVID-19 pandemic, the rapid development and deployment of novel vaccine platforms, and the amplification of vaccine-related misinformation through digital and social media, all of which have contributed to vaccine hesitancy and undermined confidence in vaccination programmes [9-13].”
The new references will be integrated in numerical order according to the Vaccines style.
2.- Methods. Line 240. A total of 483 individuals were invited to participate, of whom 398 completed the survey, yielding a response rate of 82.40%. Comment: How was the sample size determined?
We appreciate this important methodological query and have clarified this aspect in the Methods section.
Our study was designed as an observational psychometric validation conducted in a census-like fashion, inviting all eligible healthcare students undertaking clinical placements at the participating hospital during the study period. Thus:
We did not perform a formal a priori sample size calculation based on power or effect size. To address the reviewer’s concern about adequacy for factor analysis, we now explicitly justify the sample size in the text by referring to established recommendations for psychometric studies and exploratory factor analysis:
1.- With 30 items in the final questionnaire and 398 respondents, we exceed common rules-of-thumb suggesting at least 5–10 participants per item and a minimum of 200 participants for robust factor analysis.
2.- Simulation studies and methodological reviews also support sample sizes in the range of 300 or more as “good to very good” for structural validity, depending on the quality of the correlation matrix.
Accordingly, in the revised manuscript we add a sentence in limitations, such as: “All eligible students (n = 483) were invited in a census-like approach, of whom 398 completed the questionnaire (response rate 82.40%). Although no formal a priori power calculation was performed, the final sample clearly exceeds commonly recommended minimums for exploratory factor analysis (e.g., at least 5–10 participants per item and ≥200 participants overall), supporting the adequacy of the sample size for psychometric validation [37-38].”
37. Mundfrom, D.J.; Shaw, D.G.; Ke, T.L. Minimum sample size recommendations for conducting factor analyses. Int. J. Test. 2005, 5, 159-168.
38. Mokkink, L.B.; Terwee, C.B.; Patrick, D.L.; Alonso, J.; Stratford, P.W.; Knol, D.L.; Bouter, L.M.; de Vet, H.C.W. The COSMIN checklist for assessing the methodological quality of studies on measurement properties of health status measurement instruments: An international Delphi study. Qual. Life Res. 2010, 19, 539-549.
3.- “Line 195. Table. The table variables were narrated in the methodology?”
We thank the reviewer for this clarification request. We agree that all variables appearing in the tables should be fully described in the Methods section.
In the revised manuscript, we have expanded the Materials and Methods section to explicitly describe:
1.- Sociodemographic and academic variables, including:
1.a.- year of birth,
1.b.- country of study
1.c.- gender: female, male, or other. Participants who did not identify as male or female, or preferred not to disclose their gender, could select “other”,
1.d- degree programme (Medicine, Nursing, and other healthcare degrees),
1.e.- academic year,
1.f.- type of clinical placement or setting.
2.- KAP-related variables, specifying that:
2.a.- the knowledge block includes questions on awareness and understanding of different vaccines (childhood, special-risk, respiratory, hepatitis, influenza, and newer vaccines);
2.b.- the attitudes block includes items on confidence in vaccines, perceived importance and safety, and views on vaccine mandates;
2.c.- the practices block includes self-reported vaccination status for each vaccine and awareness of booster/ revaccination needs.
In the new version, in Material and Methods sections, appear (lines 95-113): “The KAP (Knowledge, Attitudes, and Practices) framework guided the categorization of the questionnaire dimensions. Specifically, dimensions classified as “knowledge” captured objective and factual information, whereas “attitudes and perceptions” reflected beliefs and subjective opinions. KAP-related variables were grouped into three blocks: (1) the knowledge block, includes questions on awareness and understanding of different vaccines (childhood, special-risk, respiratory, hepatitis, influenza, and newer vaccines); (2) the attitudes block, includes items on confidence in vaccines, perceived importance and safety, and views on vaccine mandates; and (3) the practices block includes self-reported vaccination status for each vaccine and awareness of booster/ revaccination needs. Each studied dimension corresponded to a block of questions, and a fourth block evaluated demographic aspects, which were not included in the validation process.
The demographic block included open-ended questions on year of birth, country of study, degree program, type of clinical placement or setting and the academic year. A closed-ended question on gender offered three options: female, male, or other. Participants who did not identify as male or female, or preferred not to disclose their gender, could select “other.” The remaining blocks contained single-choice closed-ended questions, except for the “Other vaccines” question, which was open-ended and allowed a free-text response.”
We have also made sure that the description of these variables in the Methods clearly corresponds to the content of the table mentioned by the reviewer (the table that presents the distribution of sociodemographic and academic characteristics, and the subsequent tables on vaccination coverage, attitudes, and practices). This ensures that every variable shown in the tables is now explicitly defined and narrated in the methodological section.
Tahnk you so much!